# Freshwater Ecosystems versus Hydropower Development: Environmental Assessments and Conservation Measures in the Transboundary Amur River Basin

**Eugene A. Simonov** [1,2,*] , **Oxana I. Nikitina** [3,*] **and Eugene G. Egidarev** [3,4]

1 Rivers without Boundaries International Coalition, Dalian 116650, China
2 Daursky Biosphere Reserve, 674480 Nizhny Tsasuchey, Russia
3 World Wide Fund for Nature (WWF-Russia), 109240 Moscow, Russia
4 Pacific Geographical Institute of the Far Eastern Branch of the Russian Academy of Sciences, 690041 Vladivostok, Russia
* Correspondence: esimonovster@gmail.com (E.A.S.); onikitina@wwf.ru (O.I.N.)

**Abstract:** Hydropower development causes a multitude of negative effects on freshwater ecosystems, and to prevent and minimize possible damage, environmental impact assessments must be conducted and optimal management scenarios designed. This paper examines the impacts of both existing and proposed hydropower development on the transboundary Amur River basin shared by Russia, China, and Mongolia, including the effectiveness of different tools and measures to minimize damage. It demonstrates that the application of various assessment and conservation tools at the proper time and in the proper sequence is the key factor in mitigating and minimizing the environmental impacts of dams. The tools considered include basin-wide assessments of hydropower impacts, the creation of protected areas on rivers threatened by dam construction, and environmental flows. The results of this work show how the initial avoidance and mitigation of hydropower impacts at early planning stages are more productive than the application of any measures during and after dam construction, that the assessment of hydropower impacts must be performed at a basin level rather than be limited to a project implementation site, and that the full spectrum of possible development scenarios should be considered. In addition, this project demonstrates that stakeholder analysis and robust public engagement are as crucial for the success of environmental assessments as scientific research is for the protection of river basins.

**Keywords:** hydropower; dam; damage; ecosystem; conservation measures; environmental assessment; environmental flows; GIS

## 1. Introduction

### 1.1. Freshwater Biodiversity and Dams

Although freshwater ecosystems cover less than 1% of the earth's surface, freshwater habitats are home to more than 10% of all known animals and about one-third of all known vertebrate species [1]. These ecosystems are also the most threatened: They are strongly affected by habitat modification, fragmentation and destruction, invasive species, overfishing, pollution, disease, climate change, etc. Freshwater ecosystem health is defined by its water quality and quantity [2], connectivity to other parts of the system and landscape, habitat condition [3], and diversity and abundance of plant and animal species [4]. Infrastructure development—especially dams—has caused a dramatic decline in

the number of connected, free-flowing rivers [5]: Currently, there are more than 50,000 large dams worldwide [6]. Some of these dams are used for hydropower, which is the largest contributor to global renewable electricity generation, supplying 16.4% of the world's electricity from all sources [7]. During the process of hydropower project development, insufficient attention is paid to impacts on the environment (such as the disruption of the natural flow regime, fragmentation of the single river ecosystem, suppression of migration paths and the changing habitats of species, greenhouse gas emissions from reservoirs, changes in sediment flow and channel processes, changes in the microclimate, transformation of biological and chemical properties of the water body) and measures to minimize these [8].

### 1.2. Assessment Hierarchy and Sequencing for Hydropower

Hydropower projects should seek to minimize their impact on natural ecosystems and ecosystem services while optimizing the project's energy generation potential. Adherence to the mitigation hierarchy from the earliest planning phase and throughout the project life cycle can achieve more sustainable hydropower generation [9].

The evolving context of available energy alternatives is implicitly present in hydropower discourse. From the beginning of the 21st century, wide recognition of the urgent need to reduce greenhouse gas (GHG) emissions was the reason for proponents of large hydro and nuclear power to promote these technologies as the most promising climate-friendly options [10] that provided unique benefits outweighing negative impacts on ecosystems and local communities [11]. With the emergence of massive wind and solar production having the potential to outcompete more expensive technologies, the same groups now describe hydropower as an important enabler and stabilizer for the mass deployment of intermittent renewable technologies [12]. Other industry-led entities, such as the Global Energy Interconnection and Development Organization, dare to propose world-wide renewable energy systems, where a unified high-voltage grid supplies load centers from remote solar, wind, and hydropower "energy bases" [13]. Therefore, an analysis of alternatives at the energy system level is no longer a formality, but a necessary first step when considering impact assessments in the energy sector.

The strategic environmental assessment (SEA) is accepted worldwide as the instrument to facilitate a more comprehensive analysis of energy development, either standing alone or in combination with integrated planning approaches, such as integrated water resources management (IWRM) or integrated river basin planning [14]. According to the Netherlands Commission for Environmental Assessment (NCEA), about 30 SEAs have been executed supporting decision-making on hydropower development, with most of those completed in Asian countries [15].

In theory, three distinct phases can be identified in the assessment and decision-making that leads to large dam development:

1.  Strategic planning to explore ways to fulfill societal needs with dams considered as one of the options;
2.  Dam prefeasibility studies, including siting studies;
3.  Dam feasibility and design after the selection of the preferred dam option [16].

In practice, only phase three is subject to an environmental assessment with formalized public participation, and hydropower project proponents will usually perform the previous phases without public consultations.

The objective is to compare environmental risks that have been mitigated at different stages of hydropower development and analyze the effectiveness of conservation measures. The hypothesis is that initial avoidance and mitigation of hydropower impacts undertaken at early planning stages are more effective in reducing damage to ecosystems in a given basin than measures undertaken during and after a dam is constructed.

The effectiveness of application of specific assessment methodologies can be judged only in the institutional and social context of its use. Governance systems as such may require formalized

assessment before decisions to invest in hydropower project are made. According to the Netherlands Commission for Environmental Assessment (NCEA), drivers (or root causes) for negative effects of dam development decisions are mostly related to deficiencies in the wider governance context such as neglect of strategic and system-based studies, favoring large dams above potentially more sustainable options without justification, and national public governance system incapable of correcting this situation due to late or insufficient involvement in the development process [16].

### 1.3. Importance of Stakeholder Engagement

Stakeholder analysis and proper engagement are crucial for environmental impact assessments (EIAs) because outcomes depend on the timely participation of stakeholders in the process, which can increase the willingness of the project developers to implement recommendations resulting from assessments. The social and cultural dimensions of dam development are widely documented and recognized [17] but are still rarely considered in assessments, especially at early stages of project identification [18]. The most neglected dimensions are an analysis of stakeholders involved in and affected by hydropower construction, and the development of tools to promote early stakeholder involvement into decision-making. In emerging economies, strategic planning is often expected to be based on objective science [19], rather than on the demands of various stakeholders [20].

While negative impacts on local communities partly resulting from incomplete assessments are well described in the literature [21], analyses of response strategies available to various stakeholder groups excluded from the assessment process are limited. One research work compared the more robust indigenous participatory mechanisms in Canada to those in Russia, where the development of the large Evenkiiskaya dam was halted partly due to active resistance from the indigenous community that was denied access to the official decision-making process [22]. Similar research on the Mekong [23] and rivers of Myanmar [24] has shown that the various facets in civil society that are denied access to decision-making may engage in developing their own disruptive strategies, including alternative assessment frameworks, and this may lead to drastic adjustments to project development processes driven by project proponents. Even companies strictly following national guidelines (and even best international practices) for public consultations during an EIA process still often experience fierce opposition from local communities, mainly when communities believe that the consultation process is being used as a tool to force hydropower projects in areas where communities do not agree in principle with dam building, or they see dams as a tool used by outside forces (e.g., central government) to impose control over territory/resources that local communities want to manage themselves [25]. Therefore, caution regarding public participation among project developers is quite understandable and underlies the strategy to make investment plans public only at the last stages of decision-making. However, a study that modeled such a process in Nepal still concluded that conflicts with local populations (inevitably embedded in dam projects) should be revealed and addressed at early stages of development and that by doing so, developers will save money and decrease risks [26]. The authors emphasized the lower implementation costs and higher chance for success when plans are changed at an early stage in the project cycle.

Many local stakeholders potentially affected by hydropower development are also unprepared to devote their resources to assess strategic planning documents. The majority of strategic planning documents, such as basin management plans, are not viewed by the public in developing/emerging economies as the real foundation for future decision-making, which is subject to the discretion of various government officials. Local actors most readily address problems that have already materialized and threatened their livelihoods, such as proposals to build a specific dam.

### 1.4. Amur River Basin: Biodiversity and Hydropower

An evaluation of the impacts of hydropower dams in the transboundary Amur basin is used to support the hypothesis that initial avoidance and mitigation of hydropower impacts can lead to better conservation outcomes. Amur is the largest transboundary river system in Northeast Asia, flowing

through Mongolia, China, and Russia and forming a natural border between China and Russia (see Figure 1). The river basin is famous for rare waterfowl, big cats, and endemic fish. The basin is still an arena for massive fish migration along the main stem and tributaries, with salmonids, sturgeon, and lamprey being three important examples [27]. The floodplains of the Amur and its tributaries create a belt of wetlands with high biodiversity value. The basin area is included in the list of Global 200 Ecoregions of the World, which are a priority for conservation efforts [28].

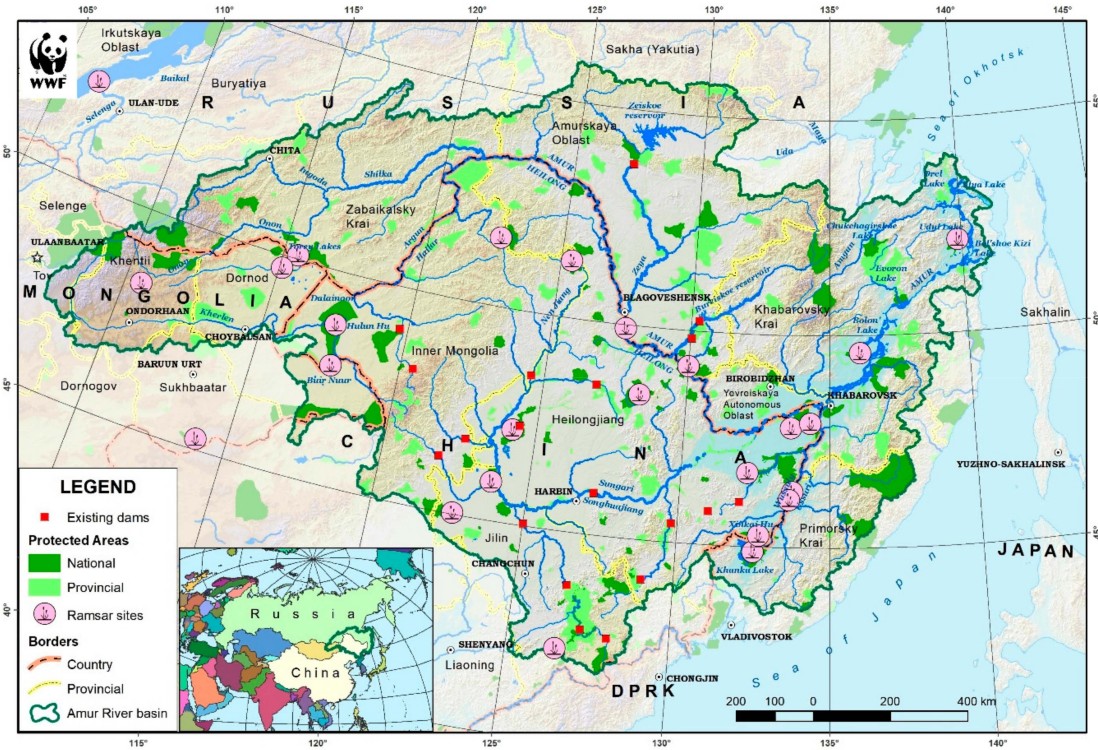

**Figure 1.** The Amur River basin and its protected areas. © WWF-Russia.

In 2019, there were approximately 100 dams in the basin, including 19 large dams, of which 3 are located in the Russian part on the Zeya and Bureya rivers. The total generation is almost 22 GWh per year, with the three large Russian dams producing 13.7 GWh per year.

In China, all the most promising sites suitable for large hydropower had already been used by the beginning of the 21st century. Recently built medium-sized dams (e.g., Hadashan on the Second Songhua River) have pursued multiple purposes, with hydropower being subordinate to irrigation, water supply, navigation, etc. Since 2015, Chinese authorities have imposed restrictions on new small hydropower construction and have started the assessment, reconstruction, and removal of smaller dams in many river basins, including the headwaters of the Second Songhua River in the Amur basin [29].

In Russia, the Amur basin was the first region where intensive hydropower construction was resumed after the economic crisis of the 1990s: The Bureya hydropower dam was put into operation in 2003 to complement the Zeya dam built in 1975, and the Lower Bureya dam started generation in 2017. The flow regime of the Zeya and Bureya have changed significantly, which has resulted in the alteration of the natural floodplain ecosystems on both rivers. This has caused a decline in typical floodplain communities [30], habitats of cranes and storks [31], and refuges for fish species [32]. Dams have become a barrier to migration, and chum salmon (keta), lamprey, and whitefish have disappeared above the dams [33]. The continuing degradation processes of the floodplain system of the Amur under the cumulative influence of the Zeya and Bureya hydropower dams has been further exacerbated below the mouth of the Songhua River, where the flow regime has undergone additional anthropogenic changes due to the construction of hydro-engineering structures in China [34].

*1.5. Initial Dialogue between CSOs and Energy Industry*

The hypothesis is that the basin-wide assessment of hydropower impacts is an effective tool in early planning when analyzing environmental costs of hydropower development and comparing possible development scenarios.

In Russia, the official environmental impact assessment methods used for hydropower projects do not allow for the analysis of complex impacts from dam construction on the ecosystem of the basin as a whole and do not compare different scenarios to optimize development [35]. For comparison, in 2011, Chinese regulations were issued for basin-wide hydropower schemes that require a basin-wide strategic assessment to precede individual dam EIAs [36]. However, the application of such assessments with meaningful public participation is yet to be fully implemented.

The lack of legal requirements and established practices for comprehensive basin-wide assessments of hydropower impacts in early 21st century Russia has become a serious obstacle to productive dialogue between the newly created state-owned Hydro-OGK hydropower company, in charge of 70% of hydropower dams (later renamed RusHydro Co.), and a wide coalition of non-governmental organizations (International Socio-Ecological Union, Greenpeace, WWF-Russia, Russian Bird Conservation Union, and other NGOs.). In the aftermath of the World Commission on Dams Report [17], the two sides sought to agree on safeguards to prevent negative impacts from new hydropower projects under specific conditions for Russia. NGO and industry experts compiled a "comprehensive list of impacts from hydropower dams and reservoirs on the environment", which identified 9 groups of problems and more than 100 specific potential impacts. An analysis of causal relationships between a multitude of effects and issues demonstrated that the majority of these were determined or strongly correlated with three fundamental environmental changes brought about by large reservoir development. These changes included (1) freshwater ecosystem fragmentation/blocking of the natural movement of biological organisms, sediments, nutrients, etc.; (2) the augmentation of natural flow dynamics of matter and energy in the river network; and (3) the creation of vast lake-like habitats serving various human activities that replaced natural river valleys. Discussions between NGOs and hydropower corporations highlighted mutual interest in identifying and ranking hydropower development options in each large river basin according to the potential severity of local and basin-wide impacts on ecosystems. This could assist both the industry and NGO community in defining long-term priorities for action and predicting and preventing possible conflicts and negative consequences. Therefore, it was suggested that the optimization of hydropower development in a large river basin first requires an assessment of impacts resulting from these three fundamental changes under different hydropower development scenarios. Such an assessment scheme was later designed by NGO experts and was initially welcomed by RusHydro management, although it was not explicitly used by the company [37].

## 2. Materials and Methods

The dialogue between RusHydro and leading civil society organizations (CSOs) held in 2007 helped to identify and introduce into the national policy debate key tools for preventing hydropower's negative impacts on nature and people who became the foundation for CSO activities in this field for the following decade. In this paper, we explore how some of those tools were applied to the Amur basin.

The tools to prevent adverse dam impacts employed in the Amur basin by civil society included the following:

1.  A strategic basin-wide assessment of hydropower impact;
2.  The creation of protected areas on the rivers that were threatened by hydropower development;
3.  Environmental flows.

The methodology underlying each of the tools designed for trial use in the Amur River basin is described in this section.

## 2.1. Strategic Basin-Wide Rapid Assessment of Hydropower Impacts

During preliminary phases of hydropower development planning, it is important to determine the scale of impacts from each potential dam and rank proposed projects and multidam development scenarios according to the degree of their environmental impacts. A basin-wide assessment of hydropower options allows for a comparison of dam sites in terms of potential effects on connectivity and downstream flow regimes. Somewhat similar basin-wide optimization approaches proposed by The Nature Conservancy (TNC) [38] and other research papers on the global footprint of dams [39] inspired the formulation of specific minimum requirements for the rapid assessment of basin-wide environmental impacts of existing and planned hydropower dams. Such an approach could yield the most significant gains if used to guide development in large basins not yet significantly altered by hydropower, and therefore the transboundary Amur basin was considered the priority among basins of Russia.

Rapid assessment methodology is focused on simple modeling of potential impacts from dams on river ecosystems and is intended to guide developers on how to choose options with the least negative environmental effects. When assessing the cumulative impacts of several dams on the ecological condition of the basin, first and foremost, we considered the following broad impact factors, which have been jointly identified as the most important by NGOs and industry experts:

1.  The alteration of flow regimes and ecosystems downstream of dams that affects the three-dimensional interaction of the river and valley;
2.  The transformation of riverine habitats in the region through their replacement by water reservoirs;
3.  The fragmentation of river networks, including the disruption of migration routes of species and material transport.

As demonstrated by previous assessments of river basins around the world, these three factors are associated with the majority of observed and predicted consequences of dam building for aquatic ecosystems, e.g., mapping the world's free-flowing rivers [5], global threats to human water security and river biodiversity [40], and restoring environmental flows by modifying dam operations [41].

To assess and compare the impacts of multiple scenarios on the river basin, three pairs of proxy indicators for impact were designed, so that each of the main factors could be expressed both in absolute and relative values.

### 2.1.1. Flow Regime Alteration and Floodplain Transformation Downstream from Dams

The most important objective should be to protect the Amur River and its floodplains, which contain the main biological resources and ecological services as well as the natural support base for the local communities of the Southern Far East. The methodology for evaluating the socioecological impacts and criteria for future dam construction should be developed based on the main concern [42]. The degree of floodplain ecosystem transformation downstream from dams reflects the consequences of altering the natural hydrological regime of a river. The index is based on the relationship between the storage volume (live volume) of a reservoir(s) and the total annual river flow volume at the dam's location, which is a consensual index used in many studies. However, for the Amur River, it is expressed not just as the percentage of the flow volume that can be withheld in the dam's reservoirs upstream of a given river's reach (index commonly known as the degree of regulation (DOR), see Reference [5]), but is further multiplied by the area of the floodplain ecosystem at a given reach. The resulting index, $Imp\_fl$ (km$^2$), is proportional to both the area of vulnerable habitats and the degree of impact in a given river reach (stretch). To characterize the basin-wide situation, it is summed across all river reaches of the basin/sub-basin ($ALT\_fl$ (km$^2$)). When related to the original unaltered floodplain area of a given river reach, it approximates the share of affected floodplains as a percent: $Imp\_fl$ (%). For each scenario, it is summed across all river reaches of the basin/sub-basin. To characterize the basin-wide alteration of floodplains, the ratio between $ALT\_fl$ (km$^2$) and total area of natural floodplains ($\sum Sfl$) in the given basin/sub-basin is used and is expressed as a percent: $100 \times \sum Imp\_fl/(\sum Sfl) = ALT\_mean$ (%).

The degree of floodplain ecosystem transformation provides an index to measure how strongly a dam or set of dams can affect the natural floodplains and their ecosystem services.

Floodplains are singled out as the most ecologically important habitat affected by flow regulation, which is valid for the Amur basin, but applying the assessment methodology in a different river ecosystem may require focusing on different key habitat types. The biota and ecosystems of the rivers in the Amur catchment are strongly dependent on the floods that are regulated by dams.

### 2.1.2. Transformation of Riverine Habitats by Reservoirs

Any reservoir is an anthropogenic feature replacing the essential socioecological landscapes: river valleys. It is assumed that the larger the surface of the water reservoirs and the greater their share in all water surfaces of the river system, then the more they transform aquatic and terrestrial ecosystems. The surface of the existing river system and the modeled surface of planned reservoirs as well as the surface of all freshwater ecosystems upstream from the dam preceding reservoir formation (during the low-flow period) is calculated. Reservoir surface areas are expressed as *Imp_res* ($km^2$). Transformation by the reservoir is calculated as the percent ratio between reservoir surface area and surface area of all aquatic ecosystems of a given sub-basin before reservoir formation *IMP_reservoir* (%).

The value of this measure, when applied to the Bureya and Zeya dams, can be seen on the impacts of fish species and stocks. The Bureya dam and Zeya dam reservoirs together occupy 3160 $km^2$, which is roughly equal to 45% of the total water surface in the Middle Amur Freshwater Ecoregion in Russia. In China, all 12 hydropower reservoirs of the basin occupy only half of that area. The Zeya and Bureya reservoirs have low-quality water, in part due to inundation by massive volumes of vegetation, soil, and peat. Before the Zeya dam construction, the composition of the fish fauna of the Upper Zeya basin in 1970 included 38 species, but by 2007 the fish fauna of the Zeya reservoir was reduced to 26 species [43]. Fish stocks of the Zeya reservoir have been seriously depressed for many years.

### 2.1.3. Blocked Sub-Basins: A Measure of Basin Fragmentation

River fragmentation indices typically measure the degree to which river networks are fragmented longitudinally by infrastructure, such as hydropower dams. Fragmentation prevents effective ecological processes that depend on longitudinal river connectivity, including the transport of organic and inorganic matter and upstream and downstream movements of aquatic and riparian species [5,39].

A simple measure of the fragmentation of the river basin is a percentage of the basin area that is cut off from the sea by dams. This study takes as a proxy measure the area (*Imp_bl* ($km^2$)) or share (*IMP_block* (%)) of the basin disconnected by dams from the sea. In other more sophisticated versions of this methodology (not presented in this paper), additional fragmentation indices have also been tested that measure the degree of partition in many disconnected sub-basins [44]. For purposes of this rapid assessment, those additional indices do not reveal new trends but reinforce the emphasis on fragmentation. In a more nuanced study with a detailed comparison of multidam scenarios, the use of such indices would be more justified.

A blocked basin index was justified as a proxy index in our analysis because the global value of the Amur Freshwater Ecoregion was primarily attributed to the abundance of diadromous fish species that suffer from disruption of migratory routes to the sea [27]. For example, upstream from the Zeya and Bureya dams, the Amur sturgeon, kaluga, keta salmon, Japanese lamprey, and other migratory species have already disappeared. Taken together, by 2011, the Zeya and Bureya dams blocked 9% of the Amur catchment area, while all of the existing dams in China blocked an additional 22–23%. This means that nearly one-third of the Amur River system has already been isolated from the sea and can no longer sustain key migratory species, e.g., diadromous fish.

For calculations, the analyzed river system was divided into reaches (stretches) limited by existing and potential dams and confluences with tributaries. Since the analysis was applied only to large dams/reservoirs, the inquiry was limited to streams having a catchment area of more than 10,000 $km^2$ (with 3–4 exceptions where large dams were planned in catchments of a smaller size).

Each stretch with a complex of natural characteristics was defined as a key basic unit in the multifactor analysis. The whole Amur basin was delineated into 214 units/stretches (see Figure S1), to which we attributed characteristics of the respective elementary sub-basins. To define stream direction, borders of watersheds, the size of planned reservoirs, hydrographic models, and data from the Shuttle Radar Topographic Mission (SRTM) [45] were used.

Satellite imagery was processed using Erdas Image software, while expert interpretation of water bodies and floodplains was performed using ArcGIS 10.5. The sources of cartographic information were Landsat Aster and Sentinel-2 satellite imagery, SRTM water body data (SWBD) [46], and vector topographic maps (1:500,000–1:100,000).

Characteristics of hydropower engineering projects have been derived from planning documents, river basin development schemes, technical documentation for dam design, etc. Annual discharge at each reach was derived from official Soviet hydrological bulletins and literature on hydropower projects and plans, e.g., hydropower master plans [47], natural resource allocation studies [48], bilateral water management schemes [49,50], and five-year plans [51]. The remaining data gaps were filled by modeling/approximation.

From all available sources, a list of 84 possible large dam locations in the Amur River basin was compiled, from which 45 projects had data sufficient to perform a basin-wide impact analysis (see Figure 2).

The characteristics were attributed to each river reach (stretch) and related sub-basin as appropriate. The three proxy indicators of impact were calculated at each river stretch and/or each sub-basin for each assessed basin-wide hydropower development scenario (see Table S1).

The values of the three indicators at the river mouth (or at the terminal point of any sub-basin/catchment) were taken as a proxy for a basin-wide score for a given scenario in this catchment. To characterize the degree of alteration from one value (following the advice of Dr. Alexander Martynov), an integrated index was also developed:

$$INT3 = \sqrt[3]{(ALT\_mean) \times (IMP\_reservoir) \times (IMP\_block)}.$$

For comparisons between basin-wide scenarios, the three proxy indicators were used as well as the integrated index calculated at the Amur River estuary.

To relate electricity production to the potential impact, all proxy indices were divided by the annual electricity production expected in a given scenario. The impact per unit of energy generation is an important measure of efficiency in terms of environmental economics [52].

All existing and possible dams and hydropower development scenarios (combinations of dams described in planning documents) in the Amur River basin were ranked according to their potential environmental impacts and efficiency relative to energy production (see Table S2).

For the sustainable development of hydropower in a given river basin, the aim is to minimize both potential impacts and their relative measure per unit of energy production. This rapid assessment methodology enables a quick initial ranking of multiple hydropower development scenarios in a large river basin according to their key potential negative impacts. Therefore, it provides developers with an opportunity to concentrate further efforts on less risky development options, while conservationists can focus on the prevention of dam construction at locations where it may lead to the greatest negative impacts for natural freshwater ecosystems.

This is the first basin-wide assessment of hydropower impacts designed for the Amur basin and the first ever suggested in Russia. The main innovation of our methodology compared to earlier assessment methods designed for other basins has been to reduce the assessment to three key variables, which, once agreed upon, do not require subjective expert inputs in further scenario assessment. This enables a credible comparison of impacts from any number of hydropower development scenarios at a basin-wide scale, including existing and proposed dams. Despite simplification, the methodology still takes into account specific features of the basin in question (e.g., using the degree of floodplain

ecosystem transformation index in place of the conventional degree of regulation index (DOR)). Our model has also been shown to have certain theoretical value by successfully relating various degrees of hydropower potential utilization to measurements of potential impacts associated with such development scenarios.

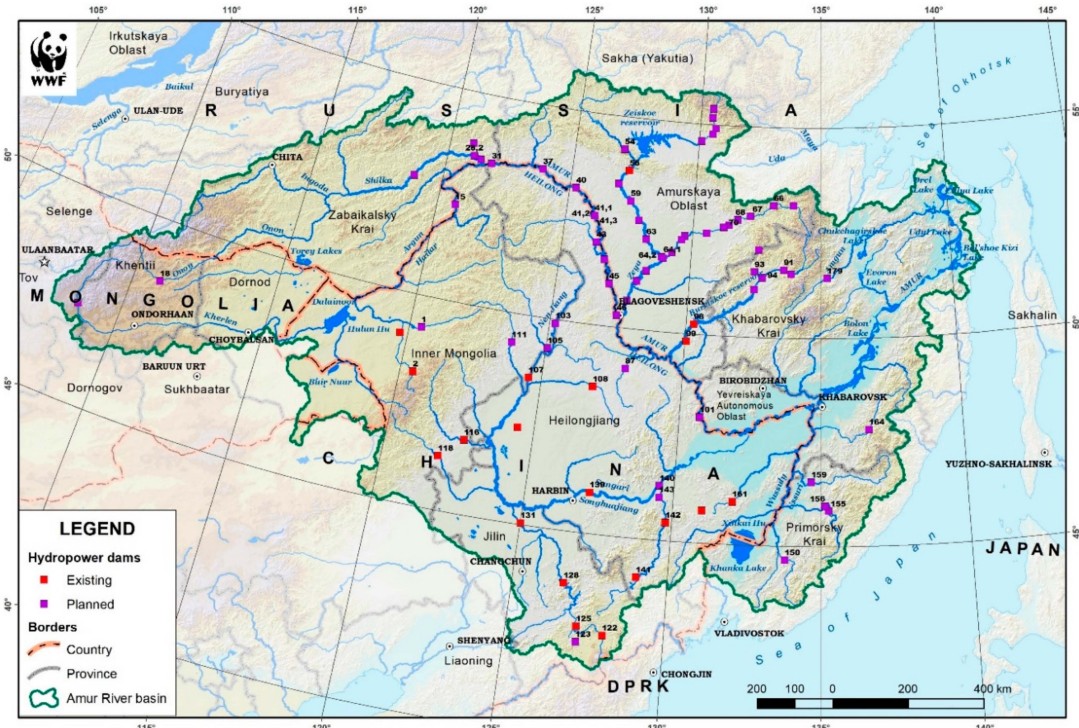

**Figure 2.** Locations of the existing and potential dams in the Amur River basin. © WWF-Russia.

### 2.2. Creation of Protected Areas at River Stretches Targeted for Hydropower Development

In the case of the Amur, the river stretches where hydropower construction is predicted to cause the most damage to the ecosystem of a given river basin are pre-emptively protected in perpetuity to constrain the future design of hydropower at such sites by legal regulations. Since sites known to be suitable for large dams are scarce and often located in river gorges with rich biodiversity, it is often possible to justify protected area development based on local natural values even without reference to its wider safeguarding function for a basin-wide river ecosystem. Since Russian legislation does not prescribe the design of specific biodiversity conservation measures as part of basin-wide water resources management planning, after the identification of critical sites, the rest of the work follows a standard methodology for the creation of protected areas.

In Russia, legally protected natural areas of different types can be established at a provincial and national level, while in China, four levels of government have such authority. In both countries, there are long-term plans for gazetting conservation areas, in which prospective protected areas should be included in order to be considered [53]. Field research by local scientists to collect data to justify the protection of a given site is usually required. In both countries, formal reports covering the scientific, economic, and management justification for a given protected area should be presented to a special commission formed by a relevant government agency. In addition, until 2019, Russia required the socioeconomic justification for protected areas (PA) establishment to be subject to an EIA procedure, which included public consultations with potentially affected stakeholders. The overall process of PA establishment could take anywhere from 2 to 10 years. However, once preservation on a natural hydrological regime and/or prohibition for massive infrastructure development is written into management regulations of a new protected area, it becomes a legal requirement. Removing or adjusting such prohibitions requires a new and lengthy procedure involving a new EIA and public

consultations. Adjustments to a protection regime happen more often in China than in Russia, but still present serious legal and reputational challenges for proponents of dam building. In China, at least one precedent of the protected area being reconfigured to give way to dam building is known. However, this has not happened in Russia.

Protected areas have been created several times since the 1990s in the Amur basin to protect the most vulnerable sites. The Norsky Strict Nature Reserve in Amurskaya Province of Russia was the first such PA (successfully created in 1998) to safeguard the confluence of the Nora and Selemdzha rivers, where development of the Dagmarskaya dam was proposed in the early 1990s. The method was pioneered by a team of local conservation biologists led by Dr. Yury Darman [54].

This team of authors (to the best of our knowledge) is the first to apply this approach to protected areas for freshwater ecosystem conservation at the basin scale, namely to establish conservation areas at potential damming sites in a systematic manner.

*2.3. Environmental Flows*

Transformation of the flow regime downstream from the dam is among the most adverse impacts of dam construction on freshwater ecosystems. One of the compensatory measures to mitigate the impacts of dam construction is the implementation of environmental flow, which is the release of a specific volume of water from the reservoir to mimic the characteristics of the natural flow variability of water and sediments [7].

There are about 200 methods for determining environmental flows [55] being used globally. Regardless of the chosen method, it is important to characterize the flow regime and its intra- and interannual variability in natural conditions. These characteristics describe the water regime to which the ecosystem has been adapted.

In Russia, guidelines for the development of reservoir operating rules indicate that reservoirs should also be used for environmental flow releases [56]. The method prescribed in "Methodological Guidelines for Developing Standards to Assess Environmental Flow" is based on preserving the freshwater ecosystem in a state where its restoration potential is not disturbed by determining the critical flow volume [57]. Environmental flow is calculated based on the difference between the values of annual runoff and the volume of allowable water withdrawal. In this study, the applicability of various methods to freshwater ecosystems of the Amur basin was assessed, but we did not seek to design another specific methodology.

## 3. Results

The description of the results is characterized, to the extent possible, both from technical outcomes of the assessments as well as from the public participation and policy processes in which they were used.

*3.1. Basin-Wide Hydropower Assessment*

### 3.1.1. Main Assessment Findings

Using the basin-wide rapid assessment tool, how potential impacts depend on the degree and pattern of hydropower development was analyzed, with the following results.

At initial phases of development in unaffected basins, individual hydropower dams with similar production may have a more than 10-fold difference in potential basin-wide environmental impacts (Figure 3).

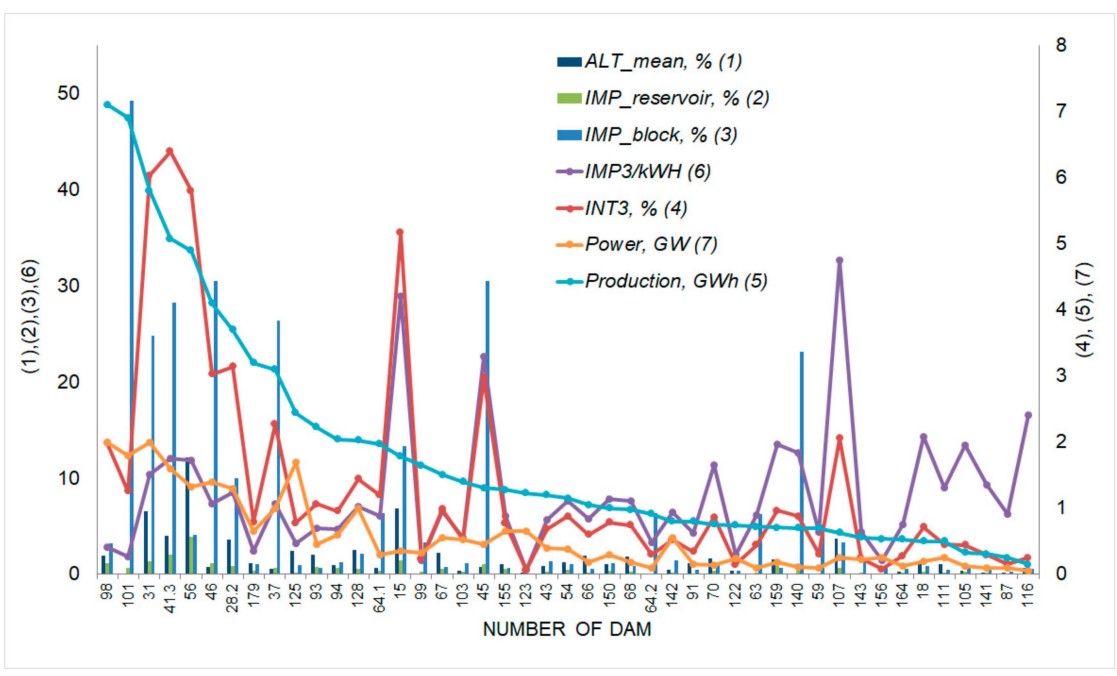

**Figure 3.** Impact of individual hydropower dams.

It was found that as the development of hydropower potential progresses, the difference between "best" and "worst" development options decreases. The scenarios selected for analysis in this paper are described in Table 1.

**Table 1.** Composition of the scenarios shown in Figures 4–6.

| | Figure 4 | | Figure 5 | | Figure 6 |
|---|---|---|---|---|---|
| Scenario 10% min | 37, 38 | Scenario Actual | 2, 56, 98, 107, 108, 116, 118, 122, 125, 128, 139, 141, 142 | Scenario actual | 2, 56, 98, 107, 108, 116, 118, 122, 125, 128, 139, 141, 142 |
| Scenario 10% max | 41.2 | Scenario Head 1 | 141, 159, 93, 155, 67 | Scenario actual, add Bureya cascade | 2, 56, 91, 93, 94, 98, 99, 107, 108, 116, 118, 122, 125, 128, 139, 141, 142 |
| Scenario 25% actual | 2, 56, 98, 107, 108, 116, 118, 122, 125, 128, 139, 141, 142 | Scenario Head 2 | 18, 68, 125, 105, 54 | Scenario actual, add low | 2, 56, 64.1, 98, 99, 107, 108, 116, 118, 122, 125, 128, 139, 141, 142 |
| Scenario 25% max | 56, 98, 15, 28.2, 70, 107, 116, 159 | Scenario Head 3 | 54, 70, 179 | Scenario actual, add Shilka | 2, 28.2, 56, 98, 107, 108, 116, 118, 122, 125, 128, 139, 141, 142 |
| Scenario 25% min | 56, 98, 91, 59, 94, 99, 93 | Scenario Head 4 | 103, 150, 94,43, 142 | Scenario actual, add Taipinggou | 2, 56, 98, 101, 107, 108, 116, 118, 122, 125, 128, 139, 141, 142 |
| Scenario 75% max | 2, 15, 31, 37, 41.2, 45, 56, 98, 101, 107, 116, 118, 122, 125, 128, 139, 141, 142 | Scenario Head 5 | 56 | Scenario actual, add mainstream | 2, 31, 41.3, 45, 56, 98, 101, 107, 108, 116, 118, 122, 125, 128, 139, 141, 142 |
| Scenario 75% min | 2, 15, 18, 43, 54, 56, 64.1, 66, 67, 68, 70, 87, 91, 93, 94, 98, 99, 103, 105, 107, 111, 116, 118, 123, 122, 125, 128, 139, 141, 142, 143, 150, 155, 156, 159, 164, 179 | Scenario Down 1 | 41.3 | | |
| Scenario 100% | 2, 15,18, 31, 37, 41.2, 43, 46, 54, 56, 59, 63, 64.1, 64.2, 66, 67, 68, 70, 87, 91, 93, 94, 98, 99, 101, 103, 105, 107, 111, 116, 118, 123, 122, 125, 128, 131, 139, 140, 141, 142, 143, 150, 155, 156, 159, 164, 179 | Scenario Down 2 | 28.2, 45 | | |

Figure 4 shows the scenarios with minimum and maximum impact, which were identified from all scenarios utilizing 10% of economic hydropower potential. In terms of the integral impact index, they differed six-fold, still showing ample potential for avoiding the most negative basin-wide impacts.

Figure 4 also shows an analysis of the actual development in the Amur basin (Scenario 25% actual), which was just below 25% of total economic hydropower potential, indicating that the same electricity production could have been achieved with a much smaller integral impact (Scenario 25% min), while the degree of impact in a scenario with maximum possible damage (Scenario 25% max) was 100% more than in Scenario 25% min, but only 10% more than in Scenario 25% actual.

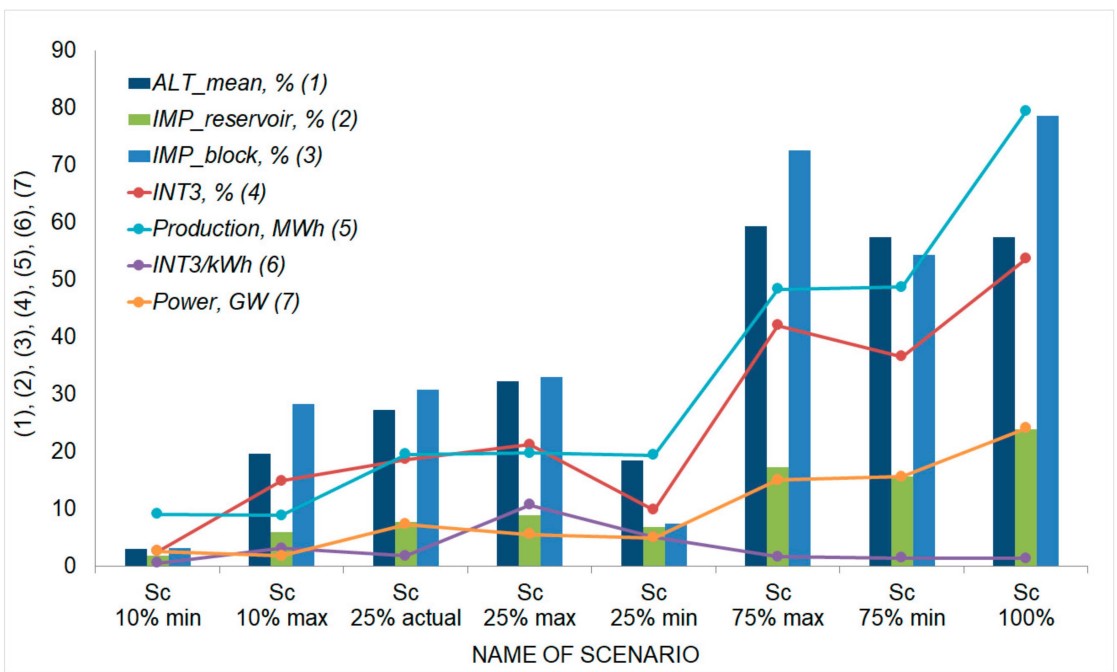

**Figure 4.** Scenario analysis and the corresponding impact due to different levels of development of the hydropower potential in the Amur basin.

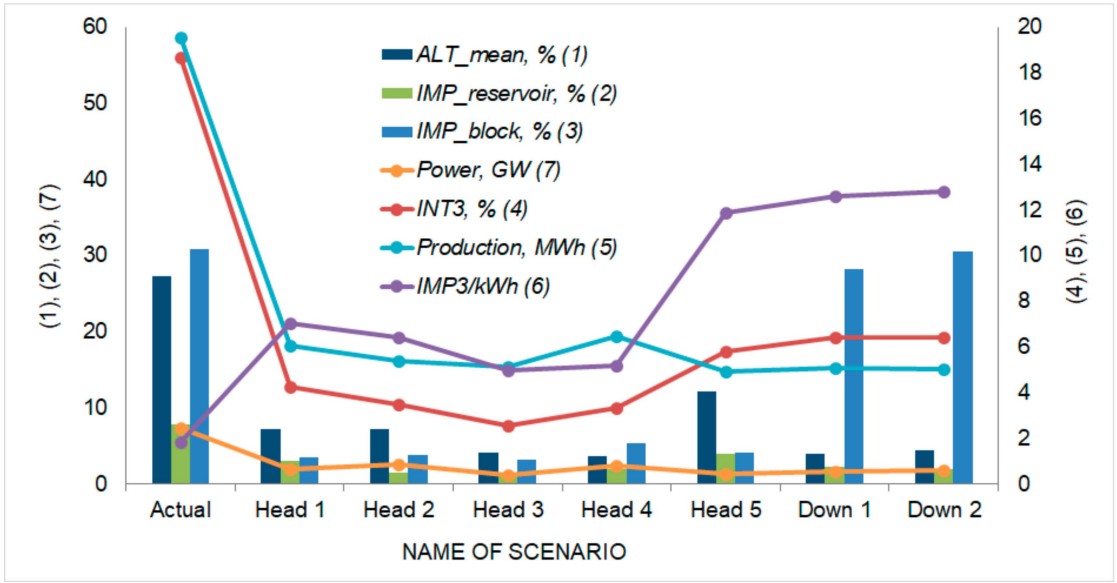

**Figure 5.** Relationship between the impact factors in scenarios when dams are located in the upper (Up) and lower (Down) parts of the Amur basin.

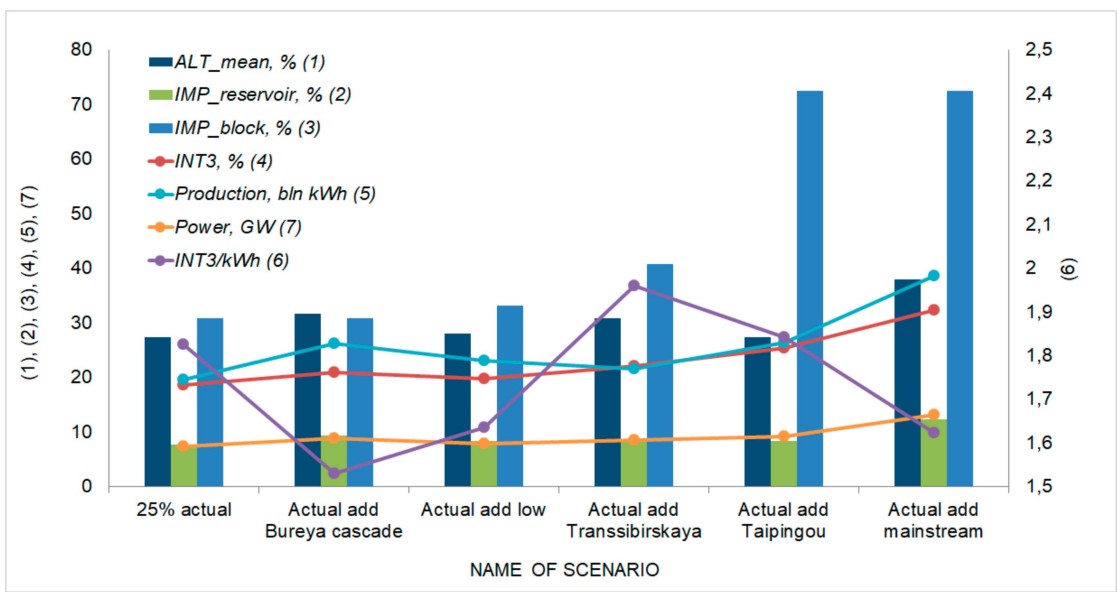

**Figure 6.** Potential scenarios of hydropower development based on the current situation.

Further modeling of scenarios with the development of 75% of technical potential resulting in maximum (Scenario 75% max) and minimal (Scenario 75% min) impact demonstrated that few choices remain between "worse" and the "worst" impacts, with only a 15% difference between the most damaging and least damaging options. Further development of "full economic potential" (Scenario 100%) led to only a 25% increase in integral impacts.

This shows that the common argument by hydropower proponents that "only 10% of hydropower potential is developed and therefore we need to develop much more" likely has an acute conflict with the notion of sustainable development because a greater proportion of basin-wide hydropower development leaves no room for "low-impact" sustainable development scenarios.

On the other hand, most impact has usually been achieved at relatively low levels of hydropower development (at least in the case of the Amur), while the additional development of dams in an already heavily affected basin brings significantly smaller additional incremental impacts. This also calls for the cautionary use of "efficiency" measures relating impact to the production of electricity, because it does not caution against excessive development in a given basin.

Dam location seems to be the leading factor defining the integral impact of hydropower development scenarios. According to this assessment methodology, "dams located downstream" are the dams that block a greater part of the overall basin. To test how this factor influences integral impact, dams were grouped in 8–9 scenarios, each having a total production of 5–6 GWh/year, with projects in a given scenario having similar "blocked basin" index values (see Figure 5). Some scenarios consisted of a single large dam on the main stem. Integral impact per unit production (*INT3*/kWh) increased for scenarios with dams located further downstream, primarily due to the increased share of the blocked basin (*IMP_block*, %), while the variation of the other two indices had no clear trend as we moved downstream. For "downstream" dams, the value of *INT3*/kWh was usually 2–2.5 times higher than for scenarios with "upstream" dams. If other aspects of fragmentation were included in the equation, dam location became an even more influential factor in defining the overall impact.

For the Amur basin, this assessment concludes that minimal additional impact is associated with the development of hydropower cascades on tributaries that already have dams. This minimizes fragmentation of the basin, interference with sediment flows, and other negative influences. Conversely, the development of dam cascades on the main stem of the Amur will lead to maximum negative impacts. The addition of even one relatively small dam (Khingansky—Taipinggou) on the main channel will lead to a sharp increase in the cumulative impacts of hydropower in the basin (see Figure 6).

However, "sustainable" development of hydropower in the Amur basin is significantly limited by the degree of cumulative impact from existing dams, which is already very close to the maximum possible impact that could be induced by any scenario with comparable electricity production.

### 3.1.2. Application of the Methodology in Business Interactions

In 2011, En+ Group/EuroSibEnergo Company, the largest private hydropower producer in Russia, announced plans to build a Transsibirskaya (Shilkinskaya) hydropower dam on the Shilka River. The company sought to diversify its business, which was primarily focused on aluminum production, and develop energy projects targeting the Chinese market and investors. The local population, as well as regional scientists and environmentalists, strongly opposed the proposed construction of a dam on the Shilka River, partly because this project had previously been proposed in the 1990s and its environmental impact assessment had shown significant negative impacts. WWF-Russia and other CSOs assisted local Russian stakeholders in developing a basin-wide campaign to communicate the potential effects and solicit feedback from civil society and local authorities in the five provinces of Russia along the Amur River [58]. To manage the conflict, the company decided to start a dialogue with civil society about sustainable hydropower options. In 2012, WWF-Russia and the En+ Group launched a research project to look into the potential for hydropower development in the Amur basin. The ultimate goal of the strategic assessment was to identify hydropower dam location options with the fewest environmental impacts for the whole Amur basin and the maximum social and economic benefit to the region. A list of 43 possible dam locations in the basin was determined, and potential environmental and socioeconomic impacts of the development on the region were analyzed.

The environmental part of the research assessed the condition of freshwater ecoregions, and one finding was that preservation of the unaltered Shilka ecoregion is essential for sustaining the ecological health of the Amur basin. The Transsibirskaya dam proposed on the Shilka River would increase the negative integral impact on the basin by 16% (see "Scenario actual, add Transsibirskaya" in Figure 6). In comparison, two additional dams on the Zeya and Bureya rivers would generate 20% more energy than the Transsibirskaya dam, but add only 4% to the integral impact on the Amur basin (see "Scenario actual, add low" in Figure 6). The assessment findings demonstrated that the increase in potential negative environmental impacts may differ by more than 10 times for development scenarios with the same additional electricity production. Dam location is the most decisive factor defining cumulative impacts on a basin scale. The best mitigation is to choose sites by using strategic assessments of the basin-wide plan and avoiding, at all costs, the development of sites that result in substantial basin-wide impacts. Existing high-impact dams severely limit the opportunities for further low-impact hydropower development and sustainable integrated water resources management.

The socioeconomic part attempted to assess the economic efficiency of the project: average prevented flood damage, macroeconomic budget efficiency, changes in employment, number of people resettled, changes in navigation conditions and turnover, losses to architectural heritage and archaeological sites, changes in fisheries, flooding, and economic flooding of objects. The socioeconomic part found that the existing Zeya and Bureya hydropower plants provide the biggest socioeconomic benefits compared to the other possible options. Among those possible dams, the Transsibirskaya dam along with the hydropower dam on the Lower Zeya and the Upper Bureya was listed as the next best option. However, the En+ Group stated that the joint assessment showed that the Transsibirskaya dam is an unbalanced/unsustainable option [44], and the company has not pursued this dam development any further.

According to the NCEA, this group is the first team of researchers and practitioners to document the process and outcomes of an SEA-like exercise carried out in real-life situations in partnership with a commercial company, without major involvement and mediation from state authorities. This makes the case study unique when compared to several other hydropower-related SEA cases that have been documented to date [11].

3.1.3. Interaction with Other Stakeholders and Further Policy Dialogue

The draft WWF&En+ Group basin-wide strategic environmental assessment report was subjected to a thorough review by local experts and representatives of various agencies. Their written comments and recommendations on the draft report were compiled and published with response matrices as an intrinsic part of the final assessment document. Besides the immediate value for the study and confirmation of its appropriateness and technical validity, such reviews played an important role in legitimizing, in the eyes of policy-makers, the results of an environmental assessment that had been initiated between an NGO and a private company and not sanctioned by authorities or academia. The company opted not to subject the economic part of the assessment to a similar review procedure, but published it with comments only from the members of the environmental assessment team.

At the same time, WWF-Russia and the En+ Group decided not to subject the report to public consultation with the local population because of the immense complexity of organizing it over the whole basin, which covers five provinces in Russia. Since the report was prepared in the Russian language, there was no formal involvement of stakeholders from China in its review either. The Rivers without Boundaries Coalition developed a summary of findings in English and shared it with relevant potential investors, including the China Yangtze Power and China Three Gorges corporations [59].

The same methodology of basin-wide hydropower assessment was applied to communicate with Russian and Chinese actors the potential impacts from a plan to develop 5–8 flood-control hydropower dams that had emerged in the aftermath of catastrophic flooding in the Amur basin in 2013. The CSOs argued that the proposed reservoirs had limited value for flow augmentation in comparison to existing natural floodplains [60], while the cumulative environmental impact from their development would be quite substantial [61]. By 2019, none of these flood control dam projects had progressed, and they are unlikely to re-emerge in the near future.

This was the first SEA-like methodology applied to basin-wide hydropower planning in Russia that received societal acceptance from various stakeholder groups, thus creating an important basis for future use of SEAs in water resources management in the country. To the best of the team's knowledge, this was also the first assessment focused on transboundary basins shared by China, and (although they did not actively participate) Chinese corporate stakeholders informed us that they used its results to inform their investment decisions.

*3.2. Establishment of the Wildlife Refuge in the Area of Proposed Hydropower Development*

In parallel with the basin-wide assessment described above, the local government, WWF-Russia, the Rivers without Boundaries Coalition, and scientists undertook assessments and negotiations to establish a wildlife refuge. In 2015, the Verkhneamursky (Upper Amur) Wildlife Refuge, with an area of 239,639 ha, was established along the Shilka, Argun, and Amur rivers [62], covering three sites previously identified as suitable for the construction of large hydropower reservoirs. One of the proposed sites was the Transsibirskaya dam, as shown in Figure 7.

The new protected area covered the potential dam building sites on both the Shilka River and Upper Amur River, and specific regulations for this wildlife refuge establishment made the future planning and development of water infrastructure illegal. To open these sites for development, the provincial government would have to undertake specific painstaking bureaucratic procedures (e.g., assessments, public consultations) to modify the legal protection regime or boundaries of the protected area, which may also serve as an additional deterrent for potential investors.

Unlike strategic environmental assessments, the procedure of creating a wildlife refuge in Russia includes mandatory public consultations with local communities. Such discussions were held in the Mogochinsky District of Zabaikalsky Province in 2015 and resulted in modification of the intended protection regime to accommodate specific traditional uses by local communities, such as hunting and fishing. Given strong opposition against dam construction on the Shilka River among local people, as well as fears of forest devastation as a consequence of the Chinese-built Amazarsky pulp mill [63], the idea to establish a vast nature reserve along the major rivers gained wide popular support.

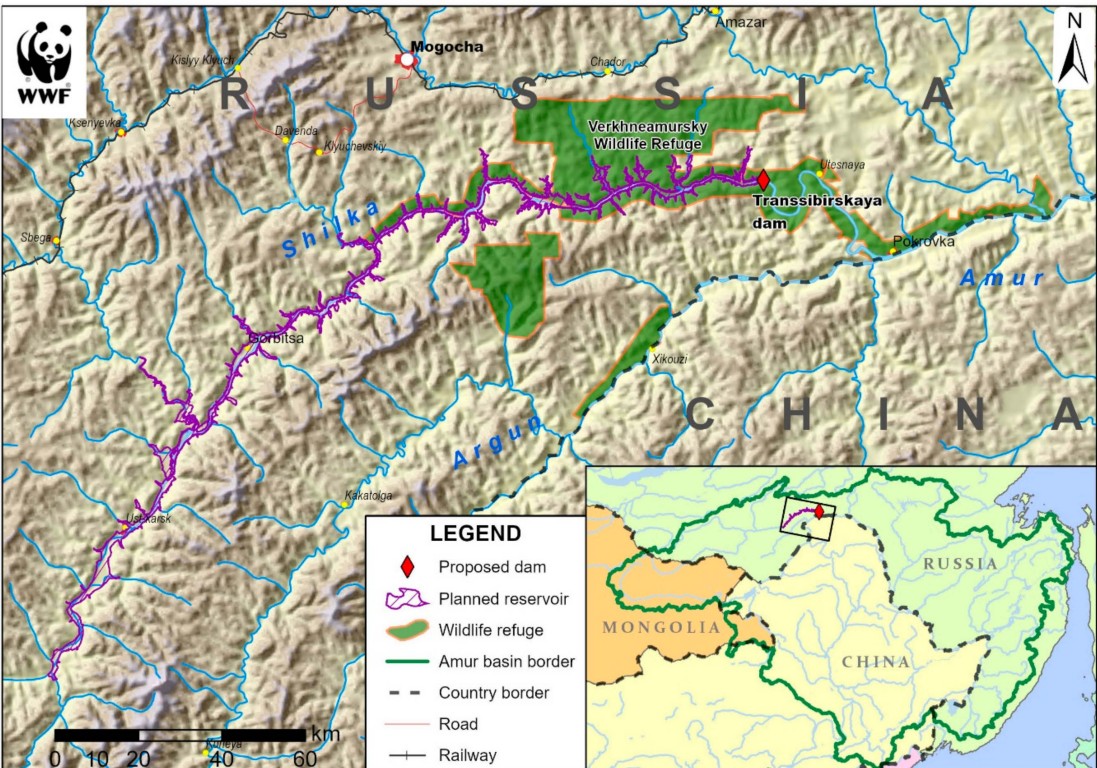

**Figure 7.** Map of the Verkhneamusky (Upper Amur) Wildlife Refuge and the potential Transsibirskaya dam site. © WWF-Russia.

### 3.3. Difficulties of Managing Environmental Flows in the Amur Basin

In the Amur basin, the main stakeholders pushing for the development of environmental releases from the already existing reservoirs are NGOs and scientists. Understanding the difficulties connected with environmental flow assessment and integration into hydropower management, NGOs have chosen a strategy to prevent initial damage instead of mitigating it after dam construction. Attempts to assess and develop environmental flows were made in 1993 [49], 2007–2010 [64], and 2013 [65]. However, as of 2019, the requirements for environmental flows from the reservoirs on the Zeya and Bureya rivers to sustain the freshwater and floodplain ecosystems below the dams had not been prescribed in the "Reservoir Operating Rules". Solving this problem is difficult due to the lack of knowledge of the biological resources of these freshwater ecosystems. In addition, for the Zeya River, there is no data on the initial conditions of the river ecosystem. This obstructs the assessment of the effectiveness of any implemented environmental flows. All these factors result in difficulties in assessing the damage caused by existing dams and developing adequate measures for its minimization and compensation.

## 4. Discussion

This section discusses how the work in the Amur basin compares to work in other basins and the possible reasons for notable variations in strategy, tactics, and outcomes. It explores aspects such as an assessment of alternatives to hydropower development, the timing of the strategic assessments and further implementation of their results, the protection of river ecosystems by establishing protected areas at undeveloped river stretches, environmental flows, and different strategies for their integration into reservoir management.

### 4.1. Amur and Baikal: Different Approaches to Alternatives

In different basins, the focus of key elements of strategic assessments could be quite different depending on local circumstances or changes in the global energy development context. In 2014–2018, members of the Rivers without Boundaries International Coalition (RwB) were involved in a basin-wide analysis of potential hydropower impacts in Lake Baikal and the Selenge River basin, shared by Mongolia and Russia. Mongolian authorities, supported by the World Bank, wanted to develop feasibility studies for two specific large dams, while the RwB members and local communities in Mongolia and Russia (as well as the World Heritage Committee) insisted that a strategic environmental assessment should first be carried out to determine the limitations for water infrastructure development in the basin and to explore available alternatives. In contrast to the analysis of alternatives in the Amur River basin, which focused on a comparison of impacts from different hydropower development scenarios, the emphasis of the analysis requested for the Lake Baikal basin was on a comparison of intended hydropower against alternative means to improve the energy supply system of Mongolia, including the deployment of solar, wind, and transboundary transmission grids, pumped storage, and other storage technologies. So far, Mongolia has agreed to conduct a regional environmental assessment for the Baikal–Selenge basin and has committed to undertake a thorough analysis of alternative technologies to the proposed hydropower dams [66]. This type of shift in focus from an analysis of dam siting to optimize hydropower development to considering alternative generating technologies at the energy system level is seen by the expert community as an important necessity (see, e.g., advice for donor governments [12], handbook on paradigm shifts in energy system planning [15], analysis of emerging trends in the Greater Mekong Subregion [67].

### 4.2. Basin-Wide Stragic Assessment: Experience in the Amur and Elsewhere

In the same decade that the WWF&En+ strategic assessment took place in Amur, many other basin-wide strategic environmental assessments (and other EIAs) have been carried out in different large transboundary basins [68]. Some of those assessments have been much more elaborate and detailed than the express analysis undertaken for Amur and were sanctioned and funded by intergovernmental organizations, governments of basin countries, donor agencies, and other relevant sources of authority. A detailed comparative study is needed in the future on the specific place and role of such SEA efforts in public education and decision-making processes [69]. Policy outcomes informed by environmental assessments vary dramatically. Shortcomings in the "Environmental Impact Report of Hydropower Development in the Upper Reaches of the Ayeyarwady River in Myanmar", performed by the Changjiang Institute of Survey, Planning, Design, and Research (CISPDR) in 2011 and subsequently highlighted by an independent review by International Rivers, contributed to the continued freezing of the controversial Myitsone dam [70]. Meanwhile, in another important case, a state-of-the-art presentation of overwhelming evidence on the negative consequences from damming the Mekong River did not stop those dams from being developed by Laos with investment from various international funders [71]. In a recent case in Myanmar, a key player, the Energy Ministry, in 2018 removed its endorsement from the final report of the most elaborate and balanced country-wide SEA, allegedly yielding to pressure from foreign firms with licenses for large hydropower development on major rivers [72]. Nevertheless, there are many examples from Vietnam, India, Pakistan, and Nepal where the recommendations of an SEA were not challenged and were partly used in decision-making. However, long-term perspectives and systemic findings of SEAs often contradict short-term interests of policy-makers and confront the shortcomings of projects already designed and invested in by developers.

The reason for the relative success of the Amur basin-wide assessment in helping to prevent further development of the most harmful hydropower projects is that it was undertaken at the earliest stages of planning, when potential project proponents were exploring project opportunities and had not yet invested heavily in a preferred option. Another important factor was the continuity of institutional knowledge and capacity building among Russian conservation NGOs and the expert community, who had, since 1990, preserved evidence on hydropower impacts that was collected during earlier

assessments. Finally, little formal involvement from governmental bureaucratic mechanisms also likely contributed to the relative ease with which the assessment outcomes were internalized by various stakeholders. Even the pledge by the company to withdraw from the Transsibirskaya dam project on the Shilka River was a voluntary action without any legal obligations or dangerous consequences.

Through the activities of the Rivers without Boundaries International Coalition, the methodology and experiences from the Amur River assessment have been shared and thoroughly discussed with experts and CSOs from China, Australia, Netherlands, Myanmar, Thailand, Nepal, and Brazil. Various elements of the methodology could be applied to designing basin-wide assessments in other regions. However, even such a simple methodology requires indices to be modified for different river basins to represent the main impact factors. New factors may also need to be added, such as water withdrawal, which was very modest in the case of the Amur basin but may be significant in other regions.

### 4.3. Conservation of River Ecosystems by Protected Areas

The movement to set aside undeveloped river stretches [73] and preserve free-flowing rivers has a long history with a growing and diverse number of precedents and supporters [74,75]. This discussion will focus only on efforts to use traditional protected areas to protect river ecosystems in East Asia. In China, extensive efforts have been undertaken by civil society to prevent the damming of the Nu River (called Thanlwin or Salween downstream in Myanmar) and to preserve the last undammed stretches of the Yangtze–Jinsha River (e.g., Tiger Leaping Gorge). Both river stretches are immediately adjacent to the "Three Parallel Rivers of Yunnan PAs" World Heritage Site, which was explicitly designed to not include riverbanks to avoid conflict with planned hydropower [76]. Nevertheless, World Heritage status has been an important factor slowing and temporarily halting hydropower development in these stretches, but there are signs it may restart in the near future since no explicit "red lines" have been drawn to limit such development in perpetuity [77].

Another example is the Upper Yangtze Rare and Endemic Fish National Nature Reserve, the last undammed stretch in the Yangtze midflow, whose boundaries were redrawn twice in 2005 and in 2011 to make way for hydropower development [78]. The first revision excluded the most valuable habitats to give way for the giant Xiangjiaba and Xiluodu hydropower projects. However, the second revision of the reserve's borders, which allowed the construction of the Xiaonanhai dam and two more dams upstream of Chongqing City, was reversed in 2015 after wide public discussion initiated by Friends of Nature and other Chinese NGOs. The presence of the Endemic Fish National Nature Reserve has been central to the argument made by NGOs and is at the heart of the reasoning in the decision announced by the Ministry of Environmental Protection to further limit dam development [79]. The Endemic Fish National Nature Reserve, while not offering absolute protection, was still an important factor in the safeguarding from damming of the last undeveloped stretch of the Yangtze, but it has limited value in terms of preserving the biodiversity of migratory fish, since it is located on a fragmented river between large dams. For example, the cascade of dams blocking a spawning habitat has led to an ongoing decline in the number of adult Chinese sturgeon in the Yangtze River and the sea, from more than 32,000 before 1981 to 2500 in 2015, and extinction in the wild is possible within the next decade [80].

On the Amur River main channel, the development of the Taipinggou National Nature Reserve in China and Verkhneamursky (Upper Amur) Wildlife Refuge in Russia has provided a certain degree of protection to the two river stretches most susceptible to damming and has played an important role in an overall multifaceted effort to preserve the main channel of the Amur as a free-flowing river.

Taking into account the timing and dynamics of investors' decision-making has been an important factor in the success of this conservation effort. The creation of such protected areas has become possible because the intention of various stakeholders to develop large hydropower at a given site was not constant over long periods, but reemerged sporadically. Attempts to dam the Amur main stem were repeated in 1986–1993, 2000, 2007, and 2016, while during the periods in between, hydropower design was not a focus of interest for any investors or politicians. In addition, in Russia, the stakeholders who are likely to benefit the most from hydropower development are not local (e.g., hydropower

companies, investors, national agencies), and the opportunity to build a dam is not seen as an essential economic opportunity by the majority of local communities or even local authorities. In contrast, in China, specific river stretches are formally assigned to large energy companies for development upon agreement with local municipalities, and this dramatically decreases the chances for successful gazetting of protected areas at such sites with the consent of local authorities.

Since freshwater biodiversity faces a severe crisis driven by impacts from large infrastructure, this approach is considered highly valuable, and work continues with CSOs globally to make business, governments, and intergovernmental bodies recognize it as an important and urgent measure to protect free-flowing rivers [29].

*4.4. Compensatory Measures to Reduce Dam Impacts on Freshwater Ecosystems*

The best tactic to protect fragile freshwater ecosystems is to avoid harmful hydropower construction by conducting a strategic environmental assessment, maintaining free-flowing rivers, and proposing careful dam siting, design, and operation. For existing dams or dams under construction, measures to minimize and compensate for their negative impacts to maintain and restore ecosystems should be implemented. Downstream effects of dams can be minimized through the provision of environmental flows and fish passages. Ecosystem restoration can be achieved through improved species, habitat, or catchment management interventions. Compensation measures implemented at different hydropower projects tend to be more effective when planned at an early stage with a basin-wide view of the situation.

On the Russian tributary of the Amur River, a program for ecosystem conservation was undertaken alongside the construction of the Lower Bureya dam (320 MW). The "Bureisky Compromise" environmental program was implemented by the hydropower company RusHydro, the local government, the UNDP implementing a project of the Global Environmental Fund (UNDP/GEF), and ecologists. The project activities were aimed at minimizing the negative impact of the Lower Bureya dam's construction on the biodiversity of the terrestrial ecosystem. In Russia, this was the first time such an environmental compensation program had been used when constructing a large dam [81]. However, the program did not include measures aimed at conserving freshwater ecosystems or species, and no study was undertaken to design environmental flow releases. As a result, the feasibility for freshwater ecosystem conservation of the Lower Bureya and Middle Amur was not assessed.

Examples of reportedly successful dam reoperation with environmental flow releases are known from several rivers of the world, including from the Three Gorges dam on the Yangtze River, China [82]. These releases, which have been provided since 2011, are successfully contributing to carp spawning [83]. However, potential improvements that could be achieved on the Yangtze via introducing environmental flows may offset only a small part of the negative impacts on freshwater biodiversity caused by extensive hydropower development. Another example of a high-resolution environmental flow assessment is the 102-MW Gulpur Hydropower Project on the Poonch River (Pakistan). The assessment resulted in decisions such as (1) reducing the dewatered section by relocating the weir closer to the powerhouse, (2) releasing environmental flows, (3) implementing a management plan for the Poonch River National Park, and (4) establishing a fish hatchery used to stock the reach downstream of the Gulpur tailrace with native fish. Additionally, the construction of two more dams on the Poonch River was cancelled due to incompatibility with the preservation of the river ecosystem in the National Park [9].

## 5. Conclusions

Hydropower and other water infrastructure development have caused a dramatic worldwide decline in the number of connected, free-flowing rivers due to haphazard planning and disregard for environmental and social values [5]. Clear limits should be placed on the allowable alteration of a large river system by water infrastructure development so that a basin can retain its vital natural processes, species diversity and abundance, vital ecosystem services, and associated cultural values.

Long-term conservation of freshwater ecosystems can only be achieved on a system scale. In most cases, a strategic assessment of hydropower has to be performed at the basin level to be successful,

rather than being limited to the project implementation site, although this has certain trade-offs in the ability to achieve all-inclusive public participation. Stakeholder analysis and proper engagement are as important for environmental assessments as scientific research. Public participation has been the key process in hydropower development, even when no formal public consultation procedure was implemented by dam project proponents.

To mitigate potential conflicts emerging at the Transsibirskaya dam, WWF-Russia and En+ Group launched a basin-wide research project at the transboundary Amur basin. The ultimate goal of the strategic assessment was to evaluate different options and identify hydropower dam location options with the least environmental impacts for the whole basin, as well as maximize the social and economic benefits to the region. To achieve this, a new methodology for the rapid environmental assessment of hydropower development in large river basins was developed. This methodology can be applied before any investments or construction takes place and is able to assess and rank options using three main factors that determine the environmental impacts of hydropower.

As a result of the project and despite the expected economic benefits from the Transsibirskaya dam, the En+ Group company stated that the joint assessment showed that a dam on the Shilka River would be an unbalanced/unsustainable option due to its environmental and social impacts. The company and its Chinese counterparts have not pursued the development of this dam any further. At the same time, development of the Lower Bureya hydropower dam proceeded smoothly without major conflicts with environmental CSOs because it was characterized by low expected impacts on the freshwater ecosystem of the Amur. Thus, the design of a common roadmap with the expectations and aspirations of both energy companies and civil society groups decreased conflict and allowed stakeholders to concentrate on identifying mutually acceptable development options.

Based on this experience, the timing and sequence of the assessment tools used is the key factor in mitigating impacts from hydropower development. Starting earlier and fully incorporating the perspectives of affected stakeholders improves the chances that optimal decisions are made and that the least negative impacts are suffered by companies, communities, and natural ecosystems. The effectiveness of initial avoidance and mitigation of hydropower impacts at early planning stages is far higher compared to applying measures after dam construction.

Applying the newly developed methodology, which includes all possible development scenarios combining existing and potential large dams in the Amur basin, it was concluded that maintenance of the free-flowing Shilka River is essential for sustaining the ecological health of the Amur basin as a whole, and the Transsibirskaya dam scored among five of the most environmentally damaging development scenarios.

Other results of the research illustrated the following:

1. Even if a hydropower scheme is carefully planned, a variety of relatively low-impact scenarios are available only for harnessing the first 10–25% of basin-wide hydropower potential. In any scenario attempting to realize a greater proportion of the technically feasible potential of the Amur basin, the expected environmental impacts grow dramatically, and the difference between the "best" and the "worst" scenarios becomes insignificant. No sustainable development options were found for utilizing the majority of basin-wide hydropower potential;

2. The main factor limiting opportunities for future sustainable development of hydropower in the Amur basin is the negative cumulative impact resulting from existing dams. If planned in an environmentally sound way, the current generating capacity could have been developed with an environmental impact two times smaller;

3. A hydropower cascade on the main stem of the Amur River would be associated with the highest basin-wide environmental impact (which is consistent with findings from many other large basins), while additional dams on tributaries already altered by hydropower are associated with the smallest additional basin-wide environmental impact.

### 5.1. Long-Term Protection of Rivers and Ecosystems

The creation of protected areas in places suitable for damming can constrain hydropower development at ecologically sensitive sites and lead to long-term protection of wild rivers and surrounding terrestrial ecosystems. The chances of success depend on the timing of the nature reserve planning vis-à-vis the hydropower project cycle and the attitudes toward hydropower held by the local population and authorities responsible for protected area development. The threat of hydropower, which certainly blocks access to many traditionally used resources, often causes local communities to proactively support nature reserve creation, despite the restrictions it imposes on land use. This approach has led to the successful establishment of three nature reserves in places suitable for hydropower dams in the Amur River basin and thus should be considered replicable at least in this large basin. Efforts to mitigate dam impacts in the Amur basin were more successful than in the Yangtze basin for a variety of reasons, such as modest prospects for hydropower development in the Amur basin, fewer obstacles in the legal system in Russia, and the difference in the hydropower development process in Russia and China. Nevertheless, the general utility of this approach has been demonstrated in the Yangtze River basin as well, albeit with less obvious long-term outcomes for basin-wide ecosystem conservation.

### 5.2. Minimization of and Compensation for Negative Impacts

For existing dams or dams under construction, measures to minimize and compensate for negative impacts and maintain and restore the affected ecosystems should be implemented. Downstream impacts can be minimized through the provision of environmental flows. Currently, requirements for environmental flow releases for the reservoirs on the Zeya and Bureya rivers have not been developed. In Russia, environmental flow releases are not implemented in practice for most reservoirs. Since the legislative framework does not force water management stakeholders to implement measures for the conservation and restoration of freshwater ecosystems, it is difficult to assess the damage caused by existing dams and to develop adequate means for its minimization and compensation. Another reason for the lack of integration of environmental flow releases is the lack of influence and interest on the part of societal groups who could potentially benefit from them. Another problem is the lack of proper monitoring and the shortage of data on the initial state of the ecosystem, without which it is difficult to assess the effectiveness of the environmental flow regime if it is implemented.

### 5.3. Building Additional Tools

The conservation methodologies developed and employed so far should be complemented in the Amur basin by detailed mapping of the distribution and connectivity between biodiversity hotspots [84]. As the first step in this work, the "Fishes of the Amur" Atlas was published [85]. The underlying database on the distribution and abundance of fish species will be used in the future to supplement basin-wide rapid assessments with a component taking into account the distribution and migration of fish species.

All conservation tools should be used in an interrelated manner in one robust planning process. To produce a conservation and development master plan for the Amur aimed at sustaining a healthy free-flowing river ecosystem, a comprehensive strategic environmental assessment of the current river basin management system and various plans for future development (including hydropower) is necessary. Of course, this arduous task is complicated by the division of our basin between China, Russia, and Mongolia, but must be undertaken to build a new "ecological civilization" together and achieve harmony between nature and humans.

**Supplementary Materials:** The following are available online at http://www.mdpi.com/2073-4441/11/8/1570/s1, Figure S1: Delineated 214 units/stretches with attributed characteristic of respective elementary sub-basins, Table S1: Calculation of the main factors of dam impact by river sections, Table S2: Primary data on hydropower dams and rivers for the production of calculations on the analyzed sections.

**Author Contributions:** Conceptualization, E.A.S.; Methodology, E.A.S. and E.G.E.; Software, E.G.E.; Validation, E.A.S., E.G.E., and O.I.N.; Formal Analysis, E.A.S.; Investigation, E.A.S. and E.G.E.; Resources, E.A.S. and E.G.E.; Data Curation, E.G.E.; Writing-Original Draft Preparation, E.A.S.; Writing-Review & Editing, O.I.N.; Visualization, E.G.E. and O.I.N.; Supervision, O.I.N.; Project Administration, O.I.N.; Funding Acquisition, E.A.S.

**Funding:** This research was funded by World Wide Fund for Nature (WWF-Russia) grant number WWF001281/RU009627-19ru/GLM, the Whitley Fund for Nature grant number 3, and the Whitley Fund for Nature supporting "Keep the Rivers Free and Wild" Project coordinated by the Rivers without Boundaries Coalition.

**Conflicts of Interest:** The authors declare no conflict of interest.

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
