# Peer review of "Freshwater Ecosystems versus Hydropower Development: Environmental Assessments and Conservation Measures in the Transboundary Amur River Basin"

_water, doi:10.3390/w11081570_

Round 1

Reviewer 1 Report

This manuscript introduced the environmental assessments and conservation measures in the transboundary Amur River Basin. The topic of this study is interesting. But some parts still need to be improved. Here are some specific comments.

(1) The literature review needs to be more critical.

(2) What is the innovation in methodology part?

(3) It’s expected to see more quantitative results for the assessment.

(4) How to ensure the proposed scenarios can cover all conditions?

(5) Please compare the results in this study with those in previous studies.

(6) What is the implication of this study for the other regions?

(7) More figures and tables are required.

Reviewer 2 Report

I can't find too much of the original research work.

Most of the manuscript is devoted to the description of the present situation and citation governmental and the other rules.

Some novel suggestion on how to improve the present situation in a balanced environmental way is not too much provided, just laments that environmental protection has not been sufficiently observed.

However generation power without very low footprints has a very strong supporting environmental effect and appropriate balace should be observed.

I have a political problem as the country of authors is not Russia, but the Russian Federation. Also, they have nowhere presented the correspondences addresses.

What means  “and 3

What means \\ ?

“World Commission on Dams (WCD)\\ Dams and Framework for Decision-Making  “

When referencing conference proceedings page numbers should be provided, when proceedings are not available the lecture identification – e.g. lecture number.

The data of the last access to websites should be provided, as the website can be changed or even removed.

When you are referencing websites, an author or authoring institution should be provided. The date of the last access should be provided as well.

http:// can be omitted as all modern browsers do not need it.

Multiple references are of no use for a reader and can substitute even a kind of plagiarism, as sometimes authors are using them without proper studies of all references used. In the case, each reference should be justified by it is used and at least short assessment provided.

For books, reports, patents, thesis and dissertations the place and country where published should be provided.

only 16 of those are made available to the public

86 (http://www.hydrosustainability.org/).             – The URL should be by hyperlink and shall be in the list of references.

13,670 million kWh per year – should be 13.67 and a proper larger unit/y

Fig 1, 2 – who owns the copyright? What about the proper reference and copyright permission?

In English text, equations and pictures, please use British standards for numbers in the text and pictures with delimiters:

1,000,000 rather than 1000000 or 1 000 000 or 1.000.000

The manuscript deserves restructuring and refocusing and after possible resubmission.

Round 2

Reviewer 2 Report

The manuscript has been revised, but not in all cases improved.

This regards to language, multiple references, style and some others.

Just to repeat the comments:

Multiple references are of no use for a reader and can substitute even a kind of plagiarism, as sometimes authors are using them without proper studies of all references used. In the case, each reference should be justified by it is used and at least short assessment provided.

For books, reports, patents, thesis and dissertations the place and country where published should be provided.

The official name of the country is Russian Federation and should be consistently used as this.

Pictures copyright permission form: It is not enough, if the ©is added. Have the authors received an official permission for using the picture? BTW Ale full refrence should be provided.

Would you avoid using "we" style in an archived paper?

In English text, equations and pictures, please use British standards for numbers in the text and pictures with delimiters:

1,000,000 rather than 1000000 or 1 000 000 or 1.000.000

The manuscript still needs further check and tide up.

Round 3

Reviewer 2 Report

The review has not been performed seriously enough.

The authors did not understand what are problems with multiple references without assessing each of them.

Nobody asked them to remove any of them.

The right way is

As demonstrated by previous assessments of river basins around the world, these three factors

237 are associated with the majority of observed and predicted consequence of dam

238 building for aquatic ecosystems [5, 40, 41].

Should be

are associated with the majority of observed and predicted the consequence of the dam

238 building for aquatic ecosystems, see, e.g. mapping the world’s free-flowing rivers [5], global threats to human water security and river biodiversity [40] and restoring Environmental Flows by Modifying Dam Operations [41].

References:

An example of the missing country is the references:

. World Wildlife Fund for Nature and The Nature Conservancy, Washington, DC 2016.

Should be

. World Wildlife Fund for Nature and The Nature Conservancy, Washington, DC, USA, 2016.

About the name of the country – it is not just formality.

Is really e.g. Tatarstan Russia or the part of the Russian Federation? Those are sensitive issues and should be followed appropriately.

The reviewer cant understand why the authors have been refusing to make appropriate small revisions.
